# Hepatitis B virus drug resistance mutations in HIV/HBV co-infected children in Windhoek, Namibia

Cynthia Raissa Tamandjou Tchuem[1¤]*, Laura Brandt[2,3], Etienne De la Rey Nel[4], Mark Fredric Cotton[4,5], Philippa Matthews[6,7], Francina Kaindjee-Tjituka[8], Wolfgang Preiser[1], Monique Ingrid Andersson[1,7,9]

1 Division of Medical Virology, Faculty of Medicine and Health Sciences, Stellenbosch University, Cape Town, South Africa, 2 International Training and Education Centre for Health, University of Washington, Windhoek, Namibia, 3 Department of Global Health, University of Washington, Seattle, Washington, United States of America, 4 Department of Paediatrics & Child Health, Faculty of Medicine and Health Sciences, Stellenbosch University, Cape Town, South Africa, 5 FAM-CRU, Department of Paediatrics & Child Health, Faculty of Medicine and Health Sciences, Stellenbosch University, Cape Town South Africa, 6 Nuffield Department of Medicine, University of Oxford, Oxford, United Kingdom, 7 Department of Microbiology, Oxford University Hospitals NHS Foundation Trust, Oxford, United Kingdom, 8 Directorate of Quality Assurance, Ministry of Health and Social Services, Windhoek, Namibia, 9 Nuffield Division of Clinical Laboratory Science, Radcliffe Department of Medicine, University of Oxford, Oxford, United Kingdom

¤ Current address: Health Economics Unit/Division, School of Public Health and Family Medicine, Faculty of Health Sciences, University of Cape Town, Cape Town, South Africa
* craissat@gmail.com

**Data Availability Statement:** All relevant data are within the manuscript and its Supporting Information files. Sensitive information that could

## Abstract

In patients who are HIV infected, hepatitis B virus (HBV) infection is an important co-morbidity. However, antiretroviral options for HIV/HBV co-infected children are limited and, at the time of this study, only included lamivudine. These children may remain on this regimen for many years until late adolescence. They are at high risk of developing HBV drug resistance and uncontrolled HBV disease. The aim of this study was to characterize HBV infection in HIV/HBV co-infected children. Known HIV-infected/HBsAg-positive children, previously exposed to lamivudine monotherapy against HBV, and their mothers were recruited at the Katutura Hospital paediatric HIV clinic in Windhoek, Namibia. Dried blood spot and serum samples were collected for HBV characterization and serological testing, respectively. Fifteen children and six mothers participated in the study. Eight of the 15 children (53.3%) tested HBV DNA positive; all eight children were on lamivudine-based ART. Lamivudine-associated resistance variants, together with immune escape mutants in the surface gene, were identified in all eight children. Resistance mutations included *rt*L80I, *rt*V173L, *rt*L180M, *rt*M204I/V and the overlapping sE164D, sW182*, sI195M and sW196LS variants. HBV strains belonged to genotypes E (6/8, 75%) and D3 (2/8, 25%). Further analysis of the HBV core promoter region revealed mutations associated with reduced expression of HBeAg protein and hepatocarcinogenesis. All six mothers, on HBV-active ART containing tenofovir and lamivudine, tested HBV DNA negative. This study confirms the importance of screening HIV-infected children for HBV and ensuring equity of drug access to effective HBV treatment if co-infected.

compromise the anonymity of the enrolled patients (age and DOB) were not included in the supporting information.

**Funding:** This work was supported by the Harry Crossley Foundation (http://www.thecrossleyfoundation.co.za/) to CRTT and the Poliomyelitis Research Foundation (https://www.prf.ac.za/) provided to MIA.

**Competing interests:** The authors have declared that no competing interests exist.

## Introduction

The effects of hepatitis B virus (HBV) infection, which is endemic in Africa, are amplified in the context of populations where HIV prevalence is high [1]. Liver disease caused by chronic viral hepatitis infection is a leading cause of death in HIV-infected individuals [2, 3]. Whilst HIV infection alone has been associated with liver fibrosis [4]. Poor immune control and higher HBV viral loads in the HIV/HBV co-infected individual, may increase susceptibility to liver damage. Furthermore, mutations in specific HBV genomic regions such as pre-S and basal core promoter (BCP) are associated with an increased risk of liver cirrhosis and hepato-cellular carcinoma (HCC) [5, 6]. The prevalence of these variants may be higher in HIV/HBV co-infected than HBV mono-infected individuals [7] and have been shown to accumulate over time [8].

HIV first-line antiretroviral therapy (ART) includes lamivudine in children <10 years and tenofovir (TDF) in those >10 years [9]. Due to the lack of ART options for children <10 years, many HIV/HBV co-infected children are exposed to lamivudine monotherapy for HBV treatment for many years. Long-term lamivudine monotherapy is associated with the evolu-tion of HBV resistance [10], leading to uncontrolled HBV replication. This is an important risk factor for developing the complications of HBV infection [11]. Selecting mutant viruses resistant to nucleos(t)ide antiviral drugs in the viral polymerase gene may cause changes in the surface gene, resulting in loss of vaccine protection [12].

Limited data are available on the prevalence of HBV infection in children in sub-Saharan Africa (SSA). Previous studies have revealed HIV/HBV co-infection prevalence rates of 0.8% in South Africa [13], and 10.4% in Zambia [14]. A 2012 study revealed an HBsAg prevalence of 8.7% among 1057 HIV-infected children < 18 years of age in Northern Namibia; a 6.5% HBsAg prevalence was observed among those aged less than one year [15]. Neither the evolu-tion of liver disease nor the molecular markers associated with liver damage in the African HIV/HBV co-infected child are well understood. This study aimed to characterize HBV in lamivudine-experienced HIV/HBV co-infected children and to assess their liver health.

## Materials and methods

### Ethical considerations

Ethical approval for this cross-sectional cohort study was obtained from the Health Research Ethics Committee at Stellenbosch University (Reference number: N13/02/022) and the Namibian Ministry of Health and Social Services. Assent was obtained from children enrolled in the study. Written informed consent was also obtained from mothers or legal guardians attending with these children.

### Study population

HIV-infected patients are screened for HBV at enrolment in HIV care in Namibia. Known HIV/HBV co-infected children and adolescents <18 years, exposed to lamivudine monother-apy for HBV, attending the Katutura Hospital paediatric HIV clinic in Windhoek were invited to take part in the study.

### HBV serology and molecular testing

Paediatric HBsAg status was confirmed using the Murex GE 34/36 Surface Antigen kit (Dia-sorin, Italy) on serum samples. Confirmed HBsAg-positive samples were analysed for anti-HBc (total), HBeAg and anti-HBe by the anti-HBc and the HBeAg/anti-HBe Diasorin kits.

Paediatric HBV DNA levels were determined using dried blood spots (DBS) on the automated COBAS® AmpliPrep/COBAS® TaqMan® HBV test V2.0 (Roche Diagnostics, Switzerland) (lower limit of quantification = 1000 IU/ml using one DBS sample). One DBS sample was eluted overnight at room temperature into a 1000 μl volume of pre-extraction buffer; from this, viral DNA was extracted and amplified by the automated system.

### HBV genotyping and mutation analysis

HBV DNA positive samples identified by the COBAS® AmpliPrep/COBAS® TaqMan® HBV test V2.0 were further analysed for Sanger Sequencing and genotyping. Viral DNA was extracted from these samples using the QIAamp® MinElute® Virus Spin kit (QIAGEN, Germany), after elution of DBS into 500 μl 1xPBS/0.05% Tween 20/0.08% sodium azide elution buffer at room temperature [16]. Genomic amplification and sequencing of HBV pol/surf, core promoter (CP) and precore (Pre-C) regions were performed as previously described [17]. Sequence chromatograms were edited, and consensus sequences were generated for each genomic region and sample. Consensus sequences (Genbank accession numbers MN651971 – MN651986) were submitted to the National Library of Medicine HBV Genotyping tool (http://www.ncbi.nlm.nih.gov/projects/genotyping/formpage.cgi) and the Stanford University HBV sequencing tool (https://hivdb.stanford.edu/HBV/HBVseq/development/HBVseq.html) for the analysis of genomic variants.

### Liver health assessment

Aspartate aminotransferase (AST) and platelet count were measured using the Architect c8000 (Abbott Laboratories, USA) and ABX Pentra XL80 (Horiba, Japan), respectively, for Aspartate aminotransferase to platelet ratio (APRI) scores calculations. An APRI score > 0.5 was suggestive of liver damage [18, 19].

## Results

### Study population

Between September 2014 and May 2015, 15 children known to be HIV/HBV co-infected (range: 8–19 years old) were recruited. Three children (aged 17–19 years) were receiving TDF-containing ART while 11 were on lamivudine-containing ART at enrolment. All 14 children had been previously exposed to lamivudine monotherapy for HBV. The treatment history of one patient was unknown.

### HBV genotyping and mutation analysis

Eight children (8/15; 53.3%) had detectable HBV DNA levels (Table 1). All were on lamivudine-based ART with full suppression of HIV replication. HBV strains grouped with genotype E (6/8; 75%) and sub-genotype D3 (2/8; 25%). Amino acid changes including $rt$L80I, $rt$V173L, $rt$L180M, $rt$V191I and $rt$M204I/V in the reverse transcriptase (RT) region of the pol gene, and $s$E164D, $s$W182*, $s$I195M and $s$W196LS in the overlapping small HBsAg (SHB) region were identified in all eight children. Additional substitutions were observed in the upper regulatory region (URR) and BCP regions: patient 2C carried the HBV URR mutant G1728A and BCP mutant G1764T/C1766G, and HBV from patients 6C and 7C carried BCP T1768A and URR C1678T mutants, respectively. Pre-C genomic changes were found in three samples: G1862T in sample 2C, and mixed populations of stop codon G1896A mutant and wild-type virus (W28*W) in samples 3C and 6C.

**Table 1. Serology and molecular analysis data of eight HIV and HBV DNA positive children and adolescents.**

| Patient ID | 2C | 3C | 6C | 7C | 8C | 10C | 13C | 17C |
|---|---|---|---|---|---|---|---|---|
| Childhood HBV vaccination | No | No | No | No | No | No | No | No |
| ART regimen at recruitment | AZT/3TC/NVP | AZT/3TC/NVP | AZT/3TC/NVP | AZT/3TC/ LPV/r | AZT/3TC/ABC/ LPV/r | AZT/3TC/NVP | AZT/3TC/EFV | AZT/3TC/ EFV |
| APRI | 0.713 | 0.481 | 0.330 | 0.321 | NA | 0.249 | NA | NA |
| HBsAg | Positive | Positive | Positive | Positive | Positive | Positive | Positive | NA |
| HBeAg | Positive | Positive | Positive | Positive | Positive | Positive | Positive | NA |
| Anti-HBe | Negative | Negative | Negative | Negative | Negative | Negative | Negative | NA |
| Anti-HBc (total) | Positive | Positive | Positive | Positive | Positive | Positive | Positive | NA |
| HBV DNA ($\log_{10}$ IU/ml) | 7.11 | 7.48 | 6.86 | 6.23 | 7.30 | 7.64 | 7.29 | 6.68 |
| HBV genotype | E | E | E | E | D3 | E | E | D3 |
| RT mutations | L180M, V191I, M204V | V173L, L180M, M204V | V173L, L180M, M204V | L180M, M204V | L80I, V173L, L180M, M204I | V173L, L180M, M204V | V173LV, L180M, M204V | L80I, M204I |
| SHB mutations | W182*, I195M | E164D, I195M | E164D, I195M | I195M | E164D | E164D, I195M | E164D, I195M | W196L |
| Pre-core mutations | G1862T | G1896A/G | G1896A/G | None | None | None | None | None |
| BCP mutations | G1728A, G1764T/ C1766G | None | T1768A | C1678T | None | None | None | None |

ID: Identification; C: Child; HBV: hepatitis B virus; ART: Antiretroviral therapy; 3TC: Lamivudine; ABC: Abacavir; AZT: Zidovudine; EFV: Efavirenz; LPVr: Ritonavir-boosted Lopinavir; NVP: Nevirapine; APRI: Aspartate aminotransferase to platelet ratio; HBsAg: hepatitis B surface antigen; HBeAg: hepatitis B envelope antigen; Anti-HBe: antibody to hepatitis B envelope antigen; Anti-HBc: antibody to hepatitis B core antigen; DNA: Deoxyribonucleic acid; IU/ml: International Units per millilitre; NA: not available; RT: reverse transcriptase, SHB: small hepatitis B surface antigen, BCP: basal core promoter.

## HBV serology

Ten of 15 paediatric samples were of adequate volume for HBV serological testing. All were reactive for HBsAg and anti-HBc (total). Seven (7/10; 70%) HBsAg-positive samples had detectable HBeAg, but anti-HBe not detected; three (3/10; 30%) did not have detectable HBeAg, of which one was anti-HBe not detected and two were anti-HBe detected.

## Liver health assessment

APRI scores were available for nine children. One HBV DNA positive child had an APRI score of 0.713 (Table 1), the other 8 were <0.5.

## Discussion

This study reports active HBV replication in 53.3% (8/15) HBV/HIV coinfected children in a paediatric cohort from Windhoek, Namibia. All HBV DNA positive children had lamivudine drug-associated HBV mutations. To our knowledge, this is the first report on the molecular characterization of HBV in paediatric HIV/HBV co-infection in Africa.

Resistance to lamivudine is best recognized in association with the *rt*M204V/I mutation [20, 21], which is also associated with decreased viral replication fitness [20], leading to the subsequent selection of the compensatory mutants *rt*L180M, *rt*V173L, and *rt*L80I/V [22, 23]. The *rt*L80I mutant was most often found in presence of *rt*M204I as opposed to *rt*M204V, and most prevalent among genotype D strains; as has been previously reported [23, 24]. Lamivudine resistance mutations detected in the RT region resulted in *s*E164D, *s*I195M, and *s*W196L/S changes in the surface gene. These are associated with reduced epitope-antibody binding, a characteristic of viral immune evasion [12]. Selection of the *s*W182 stop codon (*s*W182*) arose

from $rt$V191I. $s$W182* causes termination in the SHB gene, leading to intracellular accumulation of HBsAg protein [25]; which may increase apoptosis of hepatocytes, potentially driving hepatocarcinogenesis [25]. The presence of this variant in patient 2C, who also had an APRI score of 0.713, is in keeping with reports of liver damage associated with this mutation [26]. The risk of liver cirrhosis and HCC may also be increased in HBV-infected patients carrying BCP/Pre-C mutations, which we identified in four children. The Pre-C stop codon G1896A (W28*) mutation, observed in two children, interrupts HBeAg production [27]. The presence of both W28* and wild-type virus (W28*W) explained the HBeAg positive status noted in these two patients. It suggests the predominance of the wild-type virus over the mutant. The G1764T/C1766G BCP double mutation is enriched among cirrhotic patients [28, 29], and combined core mutations are more common in children with liver cirrhosis [30]. The accumulation of these mutants may predispose these children to severe liver disease later in life [30]. The risk of liver disease is supported by the finding of abnormal levels of biochemical markers of liver health. The APRI score was successfully determined in nine children, of whom one (patient 2C) presented with an abnormal score of 0.713. This was marginally elevated above the normal value of 0.5. Elevated APRI scores among HIV/HBV co-infected children have previously been reported elsewhere and identified as a predictor of liver disease in this population [19]. Though the diagnostic sensitivity of APRI to identify liver fibrosis is relatively low (reported to be around 47% [18]), it is worth noting that this patient also harboured HBV mutations associated with severe liver disease including $s$W182* and BCP G1764T/C1766G, and presented with a HBeAg positive status and high viraemia; these are known independent risk factors for the development of liver cirrhosis and HCC [31, 32]. The limitations of accessing liver biopsies in children and the poor access to transient elastography in this setting supports the notion that further research into non-invasive markers of liver fibrosis in children is warranted.

Controlling HBV replication is crucial for preventing liver damage, hence the necessity for effective ART in paediatric HIV/HBV co-infection. TDF or entecavir (ETV) are the preferred drugs to treat HBV [33]. At the time of this study, Namibian HIV ART guidelines recommended TDF in children older than ten years and weighing >35 kg [34]. These guidelines have been updated to prescribing TDF in children older than two years of age and weighing ≥17 kg [35], but paediatric TDF formulations have not been procured. Although decreased susceptibility to ETV may be expected in presence of HBV $rt$M204V and $rt$L180M mutants [36], doubling the dose of ETV overcomes the effects of these two mutations [37]. TDF has activity against lamivudine-drug resistant HBV [38, 39]. However, neither ETV nor paediatric TDF formulations are easily accessible in SSA. Co-formulated with elvitegravir, cobicistat, and emtricitabine, tenofovir alafenamide (TAF) has demonstrated effectiveness and safety in HIV-infected children [40] and may potentially be used for treating paediatric HIV/HBV co-infection. The drug is FDA-approved for children older than six years. Week 48 efficacy and safety data from an ongoing clinical trial in HIV-infected children suggest the potential use of the adult dose bictegravir, emtricitabine, and TAF for treating HIV/HBV co-infection in children older than six years and weighing ≥ 25kg [41]. ART regimens and formulations for children younger than six years need to be explored.

Three children (5C, 9C, and 17C), aged 18, 19 and 17 years old respectively, were treated with TDF at enrolment. Treatment with TDF was subsequent to two, seven and nine years of previous exposure to lamivudine monotherapy against HBV, respectively. Having been on a TDF-containing ART regimen for one, three and four years respectively, these children had undetectable HBV viral load at HBV testing. Due to the cross-sectional nature of this study, samples taken prior to TDF initiation were not available to determine prior existence of lamivudine-associated drug resistance mutations. Tenofovir therapy was the likely reason for the suppressed HBV viral load.

This study has identified children who require optimization of their antiviral treatment for the control of HBV infection. However, detailed imaging including ultrasonography and elastography would have provided greater breadth of data to better understand liver health in this cohort. The limitation of this study is the small sample size. This study merely serves to highlight the potential problems of HBV monotherapy where there is no access to TDF or other more potent HBV acting agents in HIV/HBV co-infected children. Larger studies are needed to ascertain the prevalence of HIV/HBV associated liver disease and importantly to determine access to potent HBV active agents for young children.

## Conclusion

HBV/HIV co-infected children, exposed to lamivudine monotherapy, are at risk of carrying scarred HBV, and developing clinically significant liver disease. The diagnosis and optimized treatment of paediatric HIV/HBV co-infection, including access to paediatric TDF formulations, should be a priority. Advocacy and funding to provide safe and effective therapy for children co-infected with HIV/HBV should be a public health priority for sSA.

## Supporting information

**S1 Dataset.**
(XLSX)

## Acknowledgments

Our appreciation goes to the staff at Medical Virology Tygerberg and at the Namibia Institute of Pathology, and the study team at the Intermediate Hospital Katutura paediatric clinic.

## Author Contributions

**Conceptualization:** Laura Brandt, Monique Ingrid Andersson.

**Data curation:** Cynthia Raissa Tamandjou Tchuem, Monique Ingrid Andersson.

**Formal analysis:** Cynthia Raissa Tamandjou Tchuem.

**Funding acquisition:** Cynthia Raissa Tamandjou Tchuem, Monique Ingrid Andersson.

**Investigation:** Cynthia Raissa Tamandjou Tchuem, Laura Brandt, Francina Kaindjee-Tjituka.

**Methodology:** Cynthia Raissa Tamandjou Tchuem, Monique Ingrid Andersson.

**Project administration:** Cynthia Raissa Tamandjou Tchuem, Monique Ingrid Andersson.

**Resources:** Laura Brandt, Wolfgang Preiser, Monique Ingrid Andersson.

**Supervision:** Laura Brandt, Wolfgang Preiser, Monique Ingrid Andersson.

**Validation:** Cynthia Raissa Tamandjou Tchuem, Monique Ingrid Andersson.

**Visualization:** Cynthia Raissa Tamandjou Tchuem.

**Writing – original draft:** Cynthia Raissa Tamandjou Tchuem, Laura Brandt.

**Writing – review & editing:** Cynthia Raissa Tamandjou Tchuem, Laura Brandt, Etienne De la Rey Nel, Mark Fredric Cotton, Philippa Matthews, Francina Kaindjee-Tjituka, Wolfgang Preiser, Monique Ingrid Andersson.

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
