## [Decision Letter · Decision Letter 0]

29 Jun 2020

PONE-D-20-15645

Drug resistance mutations in hepatitis B virus infection among HIV/HBV co-infected children in Namibia

PLOS ONE

Dear Dr. Tamandjou Tchuem,

Thank you for submitting your manuscript to PLoS ONE. We have received comments from two experts in the field. Reviewers are aware of the potential interest of your research.  However, they also agree in that your manuscript needs revision, specifically your analysis should be extended to overcome the limitations imposed by the small number of participants (see comments below)

After careful consideration, we think that your manuscript needs MAJOR REVISION in order to be considered for publication. If you are prepared to undertake the work required, we would be pleased to reconsider this decision.

We look forward to receiving your revised manuscript.

Kind regards,

Luis Menéndez-Arias, Ph. D.

Academic Editor

PLOS ONE

Journal Requirements:

Reviewers' comments:

Reviewer's Responses to Questions

**Comments to the Author**

1. Is the manuscript technically sound, and do the data support the conclusions?

Reviewer #1: Partly

Reviewer #2: Yes

2. Has the statistical analysis been performed appropriately and rigorously? 

Reviewer #1: N/A

Reviewer #2: N/A

3. Have the authors made all data underlying the findings in their manuscript fully available?

Reviewer #1: Yes

Reviewer #2: Yes

4. Is the manuscript presented in an intelligible fashion and written in standard English?

Reviewer #1: Yes

Reviewer #2: Yes

5. Review Comments to the Author

Reviewer #1: The manuscript presents data on the serologic and viral genetic characterization of HBV in HIV co-infected children recruited from a clinic in Namibia. The question under investigation is sound and methods used appropriate, and the narrative easy to follow. My major concern is the very small sample size for an epidemiologic study of this nature:

Participants were enrolled for the study for a period of about 8 months and within this time only 15 children participated. This number, for the purpose of the study is quite small, and needs to be put in context, as well as the interpretation of the findings. The question that arises is why such a small number of participants? Can this be explained by a successful MTCT HIV prevention programme in Namibia, driving down HIV/HBV coinfections, noting that the participants did not received HBV vaccines? How many children were invited and how many declined? Is the treatment uptake of HBV/HIV infected children low? Answers/discussion of these questions are important to bring the findings to be better relief. Without this, the findings remain superficial and conclusions exaggerated. Were liver function tests done, and how do the results correlate with the molecular markers of HBV pathogenesis?

Sanger sequencing was done right after HBV DNA extraction. Was amplification not done? Is there available data from deep sequencing for minority HBV drug resistant variants from elsewhere to draw parallel?

Verify: reverse transcriptase gene vs reverse transcription gene.

Reviewer #2: The paper by Tamandjou et al, shows the emergence of 3TC resistance in all children on 3TC monotherapy despite being HIV suppressed.The paper addresses an important topic of HIV/HBV coinfection especially in a setting with high prevalence of HIV. Understanding the coinfection dynamics is critical for formulation of treatment national guidelines and identifying group of patients at most risk.

This is well written paper, however, there are issues I recommend the authors to address:

Please harmonise the details of study cohort in the manuscript; results section shows different categories compared to what is seen in the abstract and methods.

The paragraph starting from line 91-94 is not clear, do authors mean one DBS sample or all DBSs tested.

It is hard to follow genotyping method. If you are referring to reference # 16, it should clearly be stated that genotyping was done as previously described and if there were changes made, they should be described.

Please expand ‘AST’ line # 103

It would be good to add extra rows to table 1 for ART and pol resistance mutations identified in these 8 children. Currently, it is not clear how mutations listed on lines #115-117 are distributed among these children.

Discussion on prevalence rates of commonly identified HBV associated resistance mutations in context of previously reported resistance mutations in Namibia would be useful; as well as persistent HBV infection in other wards HIV suppressed children. It would be useful to discuss further on 3 children with negative HBV DNA. Other than taking TDF, is there any other potential clinical independent factor for their negative HBV DNA test?

The URL for reference #33 is not functional, please revise.

6. PLOS authors have the option to publish the peer review history of their article (what does this mean?). If published, this will include your full peer review and any attached files.

Reviewer #1: No

Reviewer #2: **Yes: **Emmanuel Ndashimye

---

## [Author Response · Author response to Decision Letter 0]

6 Aug 2020

Dear reviewers,

Thank you for your very valuable comments. We have addressed each of the points raised below and edited the manuscript accordingly.

A. Reviewer #1: The manuscript presents data on the serologic and viral genetic characterization of HBV in HIV co-infected children recruited from a clinic in Namibia. The question under investigation is sound and methods used appropriate, and the narrative easy to follow. My major concern is the very small sample size for an epidemiologic study of this nature: Participants were enrolled for the study for a period of about 8 months and within this time only 15 children participated. This number, for the purpose of the study is quite small, and needs to be put in context, as well as the interpretation of the findings. 

1. The question that arises is why such a small number of participants? Can this be explained by a successful MTCT HIV prevention programme in Namibia, driving down HIV/HBV coinfections, noting that the participants did not received HBV vaccines? How many children were invited and how many declined? Is the treatment uptake of HBV/HIV infected children low? Answers/discussion of these questions are important to bring the findings to be better relief. Without this, the findings remain superficial and conclusions exaggerated. 

In response to this valuable comment, the following paragraph “The limitation of this study is the small sample size. This study merely serves to highlight the potential problems of HBV monotherapy where there is no access to tenofovir or other more potent HBV acting agents in HIV/HBV co-infected children. Larger studies are needed to ascertain the prevalence of HIV/HBV associated liver disease and importantly to determine access to potent HBV active agents for young children” has been added at lines 215 - 219 of the edited version of the manuscript. This descriptive study serves to highlight the problem of HIV/HBV co-infection in Namibia, and other African countries. The aim of the study was not to determine the prevalence of co-infection nor the prevalence of HBV drug-associated mutations. 

Were liver function tests done, and how do the results correlate with the molecular markers of HBV pathogenesis?

APRI assessment (ALT and platelets) as a non-invasive test to assess liver function was successfully performed for nine children, of whom one presented with an abnormal APRI score of 0.713. Further discussion regarding this abnormal APRI and molecular markers of HBV pathogenesis has been added in lines 175 – 186 of the revised manuscript. 

2. Sanger sequencing was done right after HBV DNA extraction. Was amplification not done? Is there available data from deep sequencing for minority HBV drug resistant variants from elsewhere to draw parallel? 

To clarify, HBV DNA amplification was performed prior Sanger sequencing. This clarification has been added in lines 100 – 105 in the revised version of the manuscript. To date, we have not performed any deep sequencing to identify minority HBV drug resistant variants. 

3. Verify: reverse transcriptase gene vs reverse transcription gene. 

The name of the gene has been amended accordingly in the manuscript.

B. Reviewer #2: The paper by Tamandjou et al, shows the emergence of 3TC resistance in all children on 3TC monotherapy despite being HIV suppressed. The paper addresses an important topic of HIV/HBV coinfection especially in a setting with high prevalence of HIV. Understanding the coinfection dynamics is critical for formulation of treatment national guidelines and identifying group of patients at most risk. This is well written paper, however, there are issues I recommend the authors to address:

1. Please harmonise the details of study cohort in the manuscript; results section shows different categories compared to what is seen in the abstract and methods.

The paragraph starting from line 91-94 is not clear, do authors mean one DBS sample or all DBSs tested. 

Thank you for pointing out this inconsistency. As suggested, the categories have been harmonised and we have clarified that one DBS sample was tested. 

2. It is hard to follow genotyping method. If you are referring to reference # 16, it should clearly be stated that genotyping was done as previously described and if there were changes made, they should be described. 

This change has been made lines 104 – 105 in the revised version of the manuscript.

3. Please expand ‘AST’ line # 103. 

AST has been expanded to “Aspartate aminotransferase (AST)”. 

4. It would be good to add extra rows to table 1 for ART and pol resistance mutations identified in these 8 children. Currently, it is not clear how mutations listed on lines #115-117 are distributed among these children. 

As suggested, we have added five rows including ART regimen at recruitment, RT mutations, SHB mutations, Pre-core mutations and BCP mutations in Table 1.

5. Discussion on prevalence rates of commonly identified HBV associated resistance mutations in context of previously reported resistance mutations in Namibia would be useful; as well as persistent HBV infection in other wards HIV suppressed children. It would be useful to discuss further on 3 children with negative HBV DNA. Other than taking TDF, is there any other potential clinical independent factor for their negative HBV DNA test? 

As added in the manuscript (lines 153 – 154 in the revised version of the manuscript), this study is a first report on the molecular characterization of drug-associated HBV resistance mutations among HIV/HBV co-infected children in Africa, reflecting the importance of this data. 

With regards to the 3 children with HBV DNA negative results due to their TDF treatment, we have added additional details regarding the duration of TDF treatment at the time of enrolment and the duration of lamivudine monotherapy against HBV prior TDF initiation in the manuscript. Due to the cross-sectional nature of the study, we could not get samples prior initiation to TDF treatment to assess whether their HBV DNA negative results were a major result of TDF treatment. This justification has been added in the discussion section of the manuscript, lines 208 to 210. Tenofovir therapy was the likely reason for the suppressed HBV viral load. 

6. The URL for reference #33 is not functional, please revise. 

A new URL has been added. This is reference #35 in the edited version of the manuscript.

---

## [Decision Letter · Decision Letter 1]

26 Aug 2020

Drug resistance mutations in hepatitis B virus infection among HIV/HBV co-infected children in Namibia

PONE-D-20-15645R1

Dear Dr. Tamandjou Tchuem,

We’re pleased to inform you that your manuscript has been judged scientifically suitable for publication and will be formally accepted for publication once it meets all outstanding technical requirements.

Kind regards,

Luis Menéndez-Arias, Ph. D.

Academic Editor

PLOS ONE

Additional Editor Comments (optional):

Reviewers' comments:

Reviewer's Responses to Questions

**Comments to the Author**

1. If the authors have adequately addressed your comments raised in a previous round of review and you feel that this manuscript is now acceptable for publication, you may indicate that here to bypass the “Comments to the Author” section, enter your conflict of interest statement in the “Confidential to Editor” section, and submit your "Accept" recommendation.

Reviewer #1: All comments have been addressed

Reviewer #2: All comments have been addressed

2. Is the manuscript technically sound, and do the data support the conclusions?

Reviewer #1: Yes

Reviewer #2: (No Response)

3. Has the statistical analysis been performed appropriately and rigorously? 

Reviewer #1: N/A

Reviewer #2: (No Response)

4. Have the authors made all data underlying the findings in their manuscript fully available?

Reviewer #1: Yes

Reviewer #2: (No Response)

5. Is the manuscript presented in an intelligible fashion and written in standard English?

Reviewer #1: Yes

Reviewer #2: (No Response)

6. Review Comments to the Author

Reviewer #1: (No Response)

Reviewer #2: (No Response)

7. PLOS authors have the option to publish the peer review history of their article (what does this mean?). If published, this will include your full peer review and any attached files.

Reviewer #1: **Yes: **Pascal O. Bessong

Reviewer #2: **Yes: **Emmanuel Ndashimye

---

## [Editor Report · Acceptance letter]

4 Sep 2020

PONE-D-20-15645R1 

Hepatitis B virus drug resistance mutations in HIV/HBV co-infected children in Windhoek, Namibia 

Dear Dr. Tamandjou Tchuem:

I'm pleased to inform you that your manuscript has been deemed suitable for publication in PLOS ONE. Congratulations! Your manuscript is now with our production department. 

Kind regards, 

on behalf of

Dr. Luis Menéndez-Arias 

Academic Editor

PLOS ONE